# DsbA-L mediated renal tubulointerstitial fibrosis in UUO mice

Xiaozhou Li[1,2], Jian Pan[1,2], Huiling Li[3], Guangdi Li [4], Xiangfeng Liu[5], Bohao Liu[1,2], Zhibiao He[1,2],
Zhengyu Peng[1,2], Hongliang Zhang[1,2], Yijian Li[6], Xudong Xiang[1,2], Xiangping Chai[1,2], Yunchang Yuan[7],
Peilin Zheng [8], Feng Liu [9,10] & Dongshan Zhang[1,2 ✉]

Recent studies have reported that upregulation of disulfide-bond A oxidoreductase-like protein (DsbA-L) prevented lipid-induced renal injury in diabetic nephropathy (DN). However, the role and regulation of proximal tubular DsbA-L for renal tubulointerstitial fibrosis (TIF) remains unclear. In current study, we found that a proximal tubules-specific DsbA-L knockout mouse (PT-DsbA-L-KO) attenuated UUO-induced TIF, renal cell apoptosis and inflammation. Mechanistically, the DsbA-L interacted with Hsp90 in mitochondria of BUMPT cells which activated the signaling of Smad3 and p53 to produce connective tissue growth factor (CTGF) and then resulted in accumulation of ECM of BUMPT cells and mouse kidney fibroblasts. In addition, the progression of TIF caused by UUO, ischemic/reperfusion (I/R), aristolochic acid, and repeated acute low-dose cisplatin was also alleviated in PT-DsbA-L-KO mice via the activation of Hsp90 /Smad3 and p53/CTGF axis. Finally, the above molecular changes were verified in the kidney biopsies from patients with obstructive nephropathy (Ob). Together, these results suggest that DsbA-L in proximal tubular cells promotes TIF via activation of the Hsp90 /Smad3 and p53/CTGF axis.

[1] Department of Emergency Medicine, The Second Xiangya Hospital, Central South University, Changsha, Hunan, People's Republic of China. [2] Emergency Medicine and Difficult Diseases Institute, Central South University, Changsha, Hunan, People's Republic of China. [3] Department of Ophthalmology, The Second Xiangya Hospital, Central South University, Changsha, Hunan, People's Republic of China. [4] Hunan Provincial Key Laboratory of Clinical Epidemiology, Xiangya School of Public Health, Central South University, Changsha, Hunan, People's Republic of China. [5] Department of Trauma center, The Second Xiangya Hospital, Central South University, Changsha, Hunan, People's Republic of China. [6] Department of Urinary Surgery, The Second Xiangya Hospital, Central South University, Changsha, Hunan, People's Republic of China. [7] Department of Chest Surgery, The Second Xiangya Hospital, Central South University, Changsha, Hunan, People's Republic of China. [8] Department of Endocrinology, Shenzhen People's Hospital, The Second Clinical Medical College of Jinan University, The First Affiliated Hospital of Southern University of Science and Technology, Shenzhen, China. [9] Department of Metabolism and Endocrinology, Metabolic Syndrome Research Center, Central South University, Changsha, Hunan, People's Republic of China. [10] Department of Pharmacology, University of Texas Health at San Antonio, San Antonio, TX, USA. ✉email: zhkidney@qq.com

Renal tubulointerstitial fibrosis (TIF) is considered the final common pathway for all kidney diseases, and is accompanied by tubular atrophy and the accumulation of extracellular matrix (ECM)[1]. A growing body of research has recognized that myofibroblasts play a key role in the production of ECM during TIF. However, recent studies have started to focus on the role of tubular epithelial cells in TIF[1,2]. Autophagy, dedifferentiation, cell cycle changes, and metabolic changes may lead to TIF via the autocrine and paracrine effects[1]. Unilateral ureteral obstruction (UUO) model, a classical TIF, not only affects the collecting ducts but also causes extensive proximal tubular degeneration[3,4]. Furthermore, a lot of evidence has shown that the proximal tubular injury caused by UUO was involved in the renal TIF[5–7]. In addition, some studies suggested that the signal pathways of both Smad3 and p53 have a profibrotic role during the progression of TIF[8–12]. However, the molecular mechanism of the proximal tubule initiating TIF remains unknown.

Disulfide-bond A oxidoreductase-like protein (DsbA-L) has been identified from the matrix of rat liver mitochondria[13]. DsbA-L has a protective role in diet-induced obesity, insulin resistance, and hepatic steatosis[14–16]. The authors of two papers have reported that overexpression of DsbA-L prevented lipid-related kidney injury in patients living with diabetic nephropathy (DN)[17,18]. Despite this, the role and regular mechanism of DsbA-L in the proximal tubule-initiating TIF remains unclear.

In the current study, the proximal tubular deletion of DsbA-L (PT-DsbA-L-KO) mice was established. Unexpectedly, we found that PT-DsbA-L-KO mice notably attenuated the UUO, ischemic/reperfusion (I/R), aristolochic acid, and repeated acute low-dose cisplatin-induced TIF. In an in vitro experiment, we found that renal fibrosis caused by transforming growth factor-β1 (TGF-β1) was ameliorated by DsbA-L short interfering RNA (siRNA), while in contrast, it was enhanced by the overexpression of DsbA-L. Mechanistically, the DsbA-L interaction with heat-shock protein 90 (Hsp90) activates the signaling pathways of Smad3 and p53 and then produces the connective tissue growth factor (CTGF) to promote TIF.

## Results

### TGF-β1-, UUO-, and Ob-induced expression of DsbA-L in BUMPT cells and in the kidney cortices of mice and patients.
First, we investigated the expression of DsbA-L in BUMPTs as well as in the kidneys of mice and human minimal change disease (MCD) patients. Immunoblot results suggested that base expression levels of DsbA-L protein were low in BUMPTs as well as in the kidney cortices of mice and human MCD patients. However, this expression was notably increased in BUMPTs treated with TGF-β1 as well as in the kidneys of UUO mice and obstructive nephropathy (Ob) patients (Fig. 1a–i). Furthermore, the immunohistochemistry or immunofluorescence staining of DsbA-L confirmed the immunoblot findings while also indicating that DsbA-L is mainly expressed in the cytoplasm of BUMPT cells or renal tubular cells of the kidney cortices of mice and human MCD patients (Fig. 1j–o).

### PT-DsbA-L-KO mice were established.
In order to clarify the role of tubular DsbA-L, we first established a mouse model of DsbA-L-KO in the proximal tubules of kidneys. The breeding protocol is described in Fig. 2a. Floxed DsbA-L alleles in male mice (DsbA-L$^{f/f}$ X$^{cre}$XY) were crossed with female phosphoenolpyruvatecarboxy kinase-cAMP-response element (PEPCK-Cre) transgenic mice (DsbA-L$^{+/+}$X$^{cre}$X$^{cre}$). Following the first-generation production, heterozygous female offsprings (DsbA-L$^{f/+}$X$^{cre}$X) were crossed with DsbA-L$^{f/f}$XY males to produce littermate mice of the proximal

tubule DsbA-L wild-type (PT-DsbA-L-WT) and PT-DsbA-L-KO (DsbA-L$^{f/f}$X$^{cre}$Y). To test genotypes, each mouse underwent three sets of PCR. The genotype of PT-DsbA-L-KO mice was characterized by (1) amplification of the 1040-bp DNA fragment floxed allele; (2) deficiency of amplification of the 915-bp DNA fragment WT allele; (3) amplification of the 370-bp DNA fragment of the *Cre* gene (see Fig. 2b, lanes 1 and 2). The lack of a *Cre* gene guaranteed the genotype of WT (PT-DsbA-L-WT) mice (see Fig. 2b, lanes 4, 5, and 7). Immunoblot analysis found that the DsbA-L protein level in the kidney cortices of PT-DsbA-L-KO mice was lower than that in PT-DsbA-L-WT, while after UUO injury, increased expression of DsbA-L in PT-DsbA-L-WT was notably suppressed in PT-DsbA-L-KO mice (see Fig. 2c, d). The immunohistochemistry result verified that DsbA-L, which had been induced by UUO in PT-DsbA-L-WT mice, tended to be ameliorated in PT-DabA-L-KO tissues, which further verified that tubular DsbA-L was deleted in this conditional KO model (see Fig. 2e).

### UUO-induced renal fibrosis, renal cell apoptosis, and infiltration of F4/80 was attenuated in PT-DsbA-L-KO mice.
The littermate mice of PT-DsbA-L-WT and PT-DsbA-L-KO were subjected to UUO for 7 days. PT-DsbA-L-KO mice markedly attenuated UUO-induced tubular dilation and atrophy (see Fig. 3a). Masson's trichome staining results indicated that the UUO-induced notable accumulation of ECM was significantly suppressed in PT-DsbA-L-KO tissues (see Fig. 3b, d). A previous study has reported that renal tubular apoptosis contributed to the accumulation of ECM[19]. In the current study, terminal deoxynucleotidyl transferase dUTP nick-end labeling (TUNEL) analysis indicated that the renal cell apoptosis caused by UUO was reduced in PT-DsbA-L-KO tissues (see Fig. 3c, f). As we know, inflammation played a critical role in the progression of renal fibrosis. The staining of F4/80 indicated that UUO-induced infiltration of macrophage into PT-DsbA-L mice kidney tissues was notably ameliorated in PT-DsbA-L-KO tissues (see Fig. 3d, g). Collectively, these data suggest that tubular DsbA-L-mediated renal fibrosis progression is associated with the promotion of apoptosis and inflammation.

### PT-DsbA-L-KO mice attenuated UUO-induced expression of collagen I and III (Col I&III) and α-SMA.
In order to further confirm whether renal fibrosis induced by UUO was reduced in PT-DsbA-L-KO tissues, we detected the expression of Col I&III and α-smooth muscle actin (α-SMA). Immunoblot results demonstrated that UUO-induced expression of both Col I&III and α-SMA was markedly attenuated in PT-DsbA-L-KO tissues (see Fig. 4a–e). Immunohistochemical staining results further confirmed the findings of the immunoblot (see Fig. 4g–i). These data supplied strong evidence to support the conclusion that PT-DsbA-L-KO mice may attenuate the renal fibrosis caused by UUO.

### DsbA-L mediated the expression of Col I&III and vimentin caused by TGF-β1 in BUMPT cells.
To verify the in vivo finding, BUMPT cells were firstly transfected with DsbA-L siRNA either with or without TGF-β1. The DsbA-L siRNA treatment notably attenuated the TGF-β1-induced expression of DsbA-L, Col I&III, and vimentin (see Fig. 5a), which was confirmed by the gray-level analysis (see Fig. 5b–e). Second, DsbA-L plasmids were also transfected into BUMPT cells and then treated with or without TGF-β1. Results demonstrated that TGF-β1-induced expression of DsbA-L, Col I&III, and vimentin was enhanced by the overexpression of DsbA-L (see Fig. 5f–j). Data were consistent with the in vivo finding.

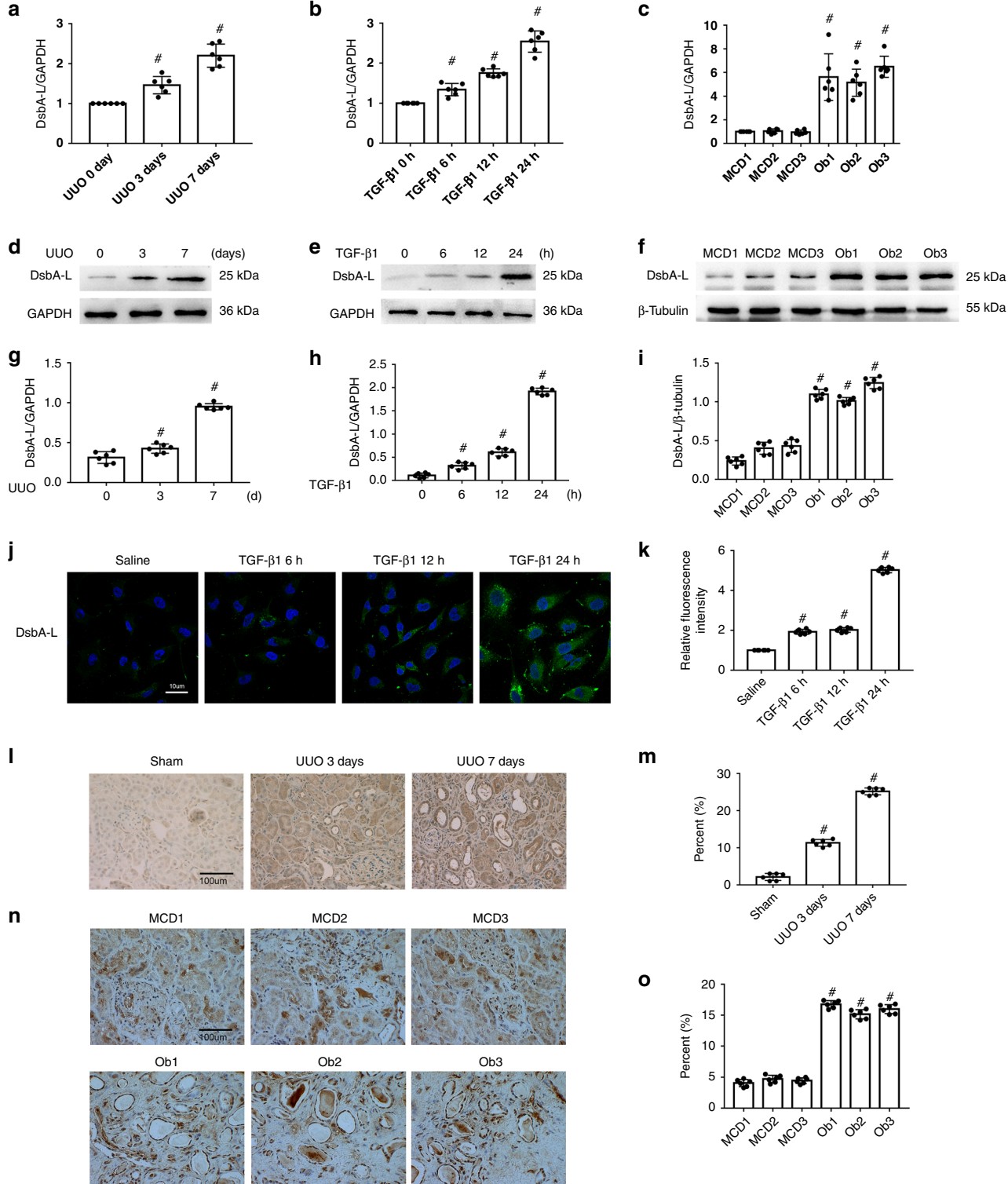

**Fig. 1 Induction of DsbA-L in BUMPT cells treated by TGF-β1 and the kidneys of UUO mice and Ob patients. a–f** RT-qPCR and immunoblot analysis of DsbA-L and GAPDH in BUMPT cells treated with or without TGF-β1 and the kidneys of UUO mice and Ob patients. **g–i** Analysis of the grayscale image between them. **j, l, n** Immunofluorescence or immunohistochemical staining of DsbA-L in BUMPT cells treated with or without TGF-β1 as well as in the kidneys of both UUO mice and Ob patients. **k, m, o** Quantification of DsbA-L staining. Original magnification ×400. Scale bar: 100 μM. Data are expressed as means ± s.d. ($n = 6$). #$P < 0.05$: versus saline group, sham group, or MCD group. Each experiment (**j, l, n**) was repeated six times independently with similar results. **a–c, g–i, k, m, o** indicate the statistical Student's $t$ test used (means ± s.d., $n = 6$, $P < 0.05$).

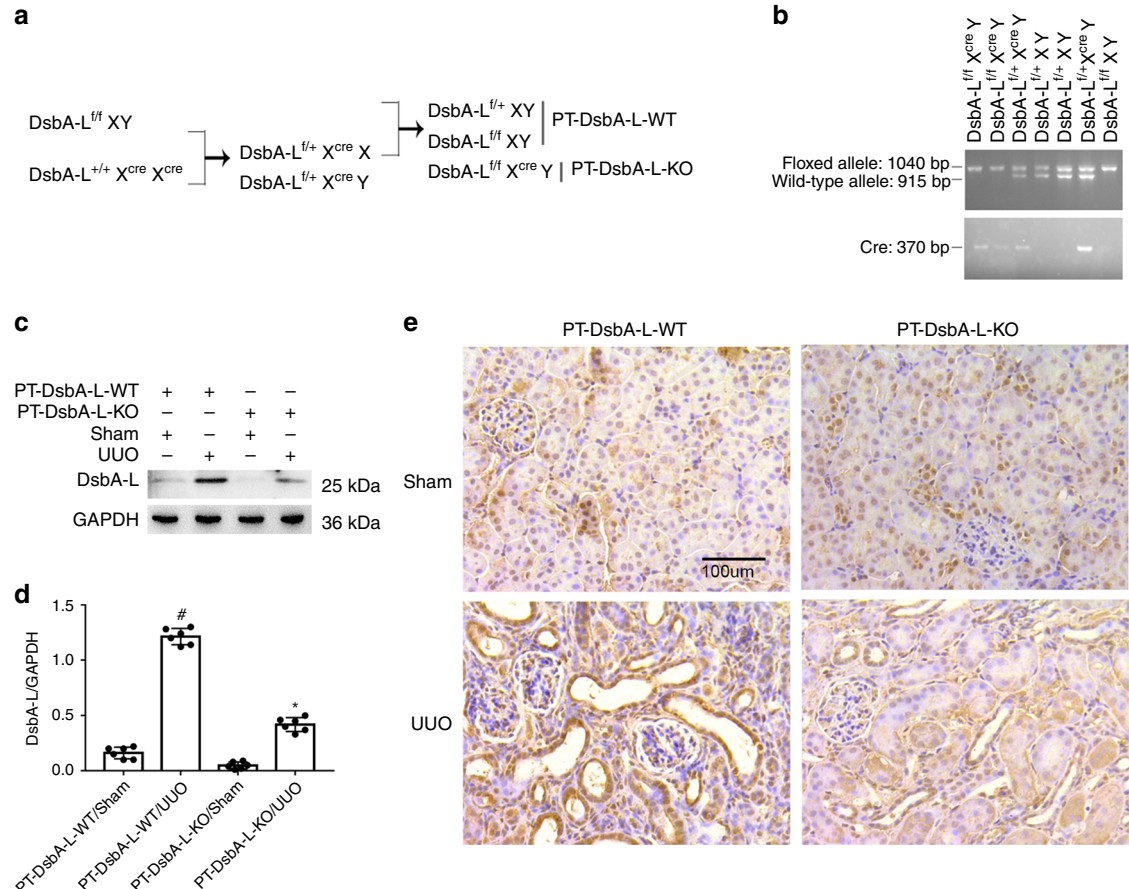

**Fig. 2 Generation and features of the PT-DsbA-L-KO mouse model. a** Breeding procedure for the creation of PT-DsbA-L-KO mice. **b** PCR-based genotyping of wild-type and floxed alleles of DsbA-L and PEPCK-Cre allele. **c** Cortices and outer medulla of kidneys from PT-DsbA-L-KO and PT-DsbA-L-WT littermate mice following UUO injury were collected for immunoblot analysis of DsbA-L and GAPDH. **d** Analysis of the grayscale image between them. **e** Immunohistochemical staining of DsbA-L in the kidney cortical tissues of wild-type and DsbA-L-KO mice following UUO injury. Original magnification ×400. Scale bar: 100 μM. Data are expressed as means ± s.d. ($n = 6$). #$P < 0.05$ versus sham group. *$P < 0.05$ versus PT-DsbA-L-WT with UUO group. Each experiment (**c**, **e**) was repeated six times independently with similar results. **d** indicate the statistical Student's $t$ test used (means ± s.d., $n = 6$, $P < 0.05$).

**Interaction of DsbA-L and Hsp90 in BUMPT cells treated with TGF-β1 as well as in the kidneys of UUO mice and Ob patients**. Previous results suggested that Hsp90-mediated renal fibrosis was induced by TGF-β1 treatment[20]. We therefore hypothesized that DsbA-L interacted with Hsp90 to regulate the renal fibrosis. Immunoprecipitation (IP) methods were used in the current experiment. Whole-cell and kidney lysates were extracted. The results demonstrated that anti-Hsp90 precipitated both Hsp90 and DsbA-L (see Fig. 6a), while anti-DsbA-L pulled down both DsbA-L and Hsp90 in control BUMPT cells, sham mice, and MCD patients (see Fig. 6a). The mitochondrial lysate of the cell and kidney was further extracted for co-IP of DabA-L and Hsp90. The data also confirmed the above findings (see Fig. 6b). The co-IP of DabA-L and Hsp90 was consistently examined at 24 h and 7 days after the TGF-β1 treatment of UUO mice and Ob patients (see Fig. 6a, b). The structure of Hsp90 contains nucleotide-binding domain/ATPase domain (20–220, amino-terminal domain (NTD)), link (220–275), structure-binding domain (275–545, middle domain (MD)), and helical region for dimerization (545–690, carboxyl-terminal domain (CTD)) (see Fig. 6c).

Based on the structure of Hsp90, we predicted that the two models of DsbA-L would interact with the NTD or CTD domain of Hsp90 using a software (see Fig. 6d). Interestingly, the prediction result indicated that one of Hsp90 isoforms, heat-shock protein Hsp90β, the so-called Hsp90 ab1 (gene ID: 15516), was the interaction partner of DsbA-L. To further investigate how the

domains of DsbA-L interacted with Hsp90, we constructed five HA-Tag plasmids with the doxycycline-inducible Tet-On promoter system, including NTD, NTD + Link, NTD + Link + MD, NTD + Link + MD + CTD, and full genes (see Fig. 6c). Anti-HA was used to pull down DsbA-L in NDTNTD + Link + MD + CTD and full gene groups, but not in groups of NTD, NTD + Link, and NTD + Link + MD plasmids, indicating that DsbA-L interacted with the CTD region of Hsp90β (see Fig. 6c).

In addition, we want to know which specific regions of CTD and Hsp90β interacted with DabA-L. Therefore, the HA-Tag CTD mutation of Hsp90β plasmids (Δ546–552; Δ667–668;681–684; Δ546–552;667–668;681–684) was transfected into BUMPT cells, while anti-HA only pulled down CTD Δ546–552, but not CTD Δ667–668;681–684 and CTDΔ546–552;667–668;681–684 (see Fig. 6e). This further demonstrated that DsbA-L interacted with the 667–668 and 681–684 region of CTD of Hsp90β (see Fig. 6f). The model and video of DsbA-L interacted with CTD of Hsp90β and simulated the above findings (see Fig. 6g and Supplementary video). Collectively, data indicated that DsbA-L interacted with the CTD region of Hsp90β.

**Co-localization of DsbA-L or Hsp90 with the mitochondria, DsbA-L and Hsp90 in BUMPT cells treated with TGF-β1 as well as in kidneys of UUO mice and Ob patients**. In order to further confirm the IP results, a co-localization assay was

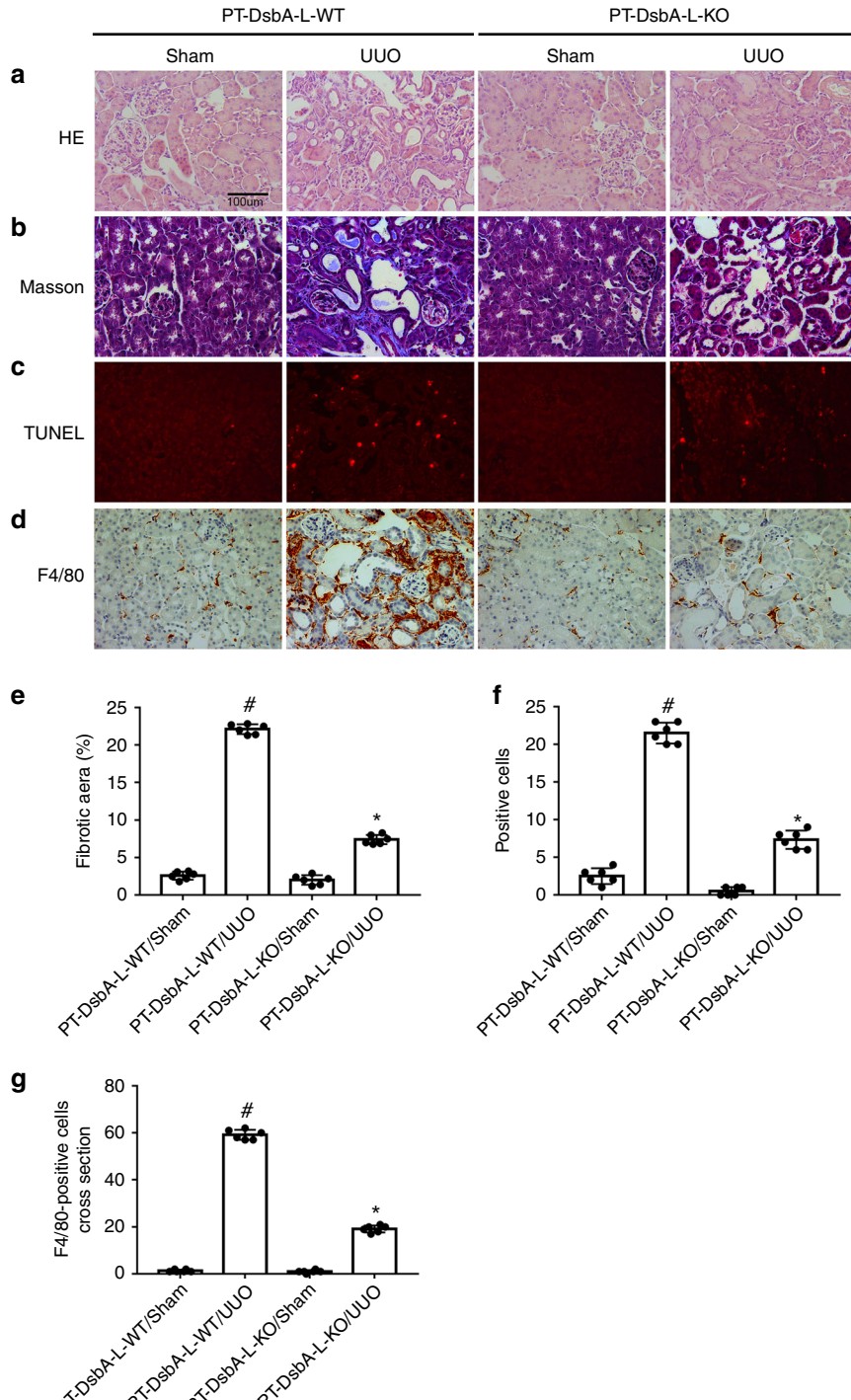

**Fig. 3 Amelioration of UUO-induced renal fibrosis and tubular cell apoptosis in PT-DsbA-L-KO mice.** The left ureter of PT-DsbA-L-KO and PT-DsbA-L-WT littermate mice was ligated for 7 days to establish a UUO model. **a** Hematoxylin and eosin staining. **b** Representative Masson's trichrome staining. **c** Representative sections of TUNEL-positive cells. **d** Immunohistochemistry of F4/80 staining. **e** Quantification of tubulointerstitial fibrosis in the kidney cortex. **f** The number of TUNEL-positive cells. **g** Quantification of the F4/80-positive cells in the kidney cortex. Original magnification ×400. Scale bar: 100 μM. Data are expressed as means ± s.d. (*n* = 6). #*P* < 0.05 versus sham group. *\**P* < 0.05 versus PT-DsbA-L-WT with the UUO group. Each experiment (**a**–**d**) was repeated six times independently with similar results. **e**–**g** indicate the statistical Student's *t* test used (means ± s.d., *n* = 6, *P* < 0.05).

performed. A previous study reported that DsbA-L was localized in the matrix of mitochondria of liver cells[13,15,21]. In this current study, an immunofluorescence confocal microscopy found that DsbA-L localizes to the mitochondria in BUMPT cells and in the kidneys of both sham mice and MCD patients, which was enhanced in BUMPT cells treated with TGF-β1 as well as in the

kidneys of UUO mice and Ob patients (see Supplementary Fig. S1A–C). The previous study demonstrated that Hsp90 was mainly localized in the mitochondria of tumor cells[22,23], and a recent study reported that while Hsp90 was induced by UUO injury, its localization remains unknown. The immuno-fluorescence confocal microscopy showed that localization of

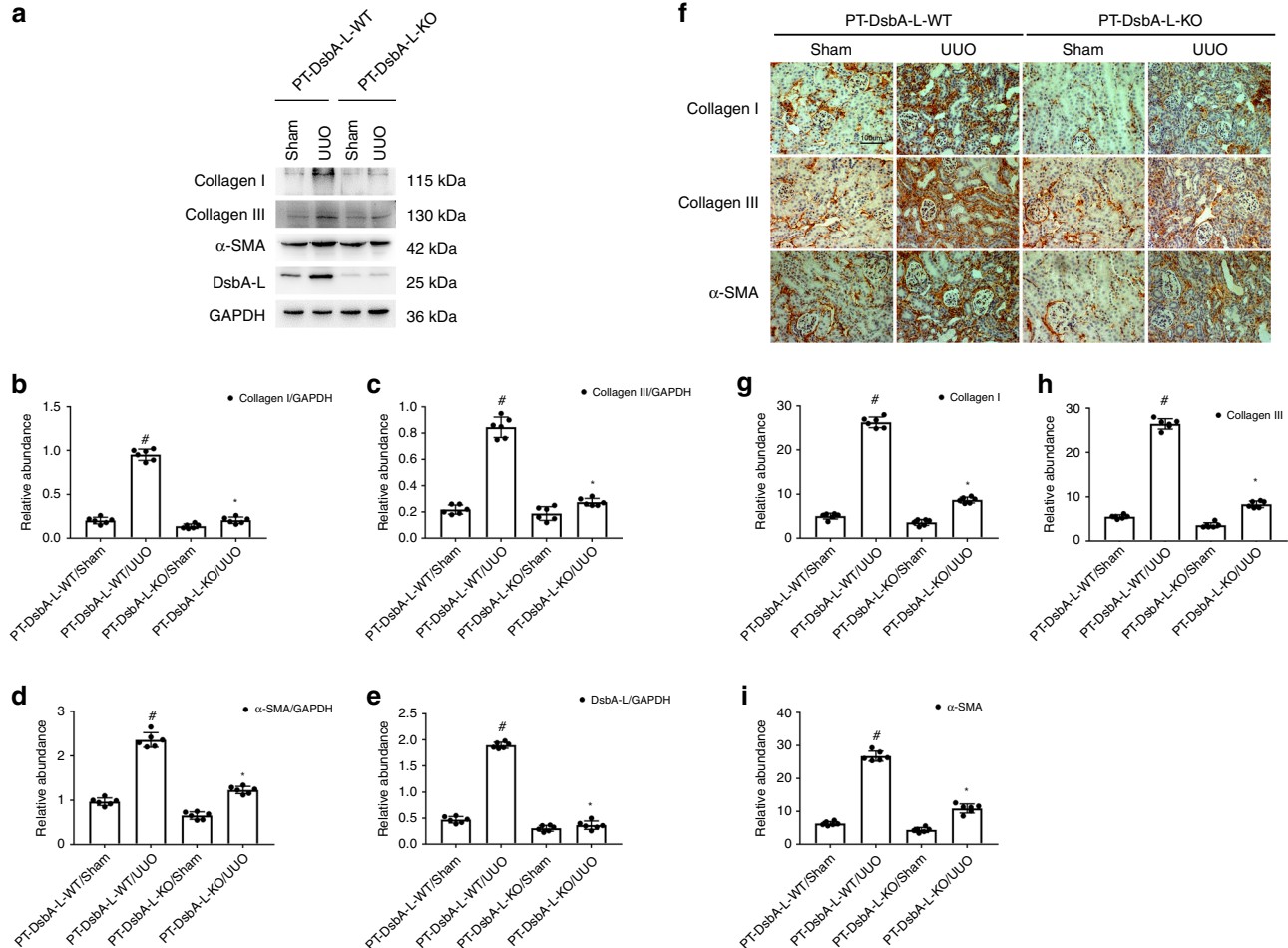

**Fig. 4 Attenuation of UUO-induced expression of Col I&III and α-SMA in PT-DsbA-L-KO mice.** The left ureter of PT-DsbA-L-KO and PT-DsbA-L-WT littermate mice was ligated for 7 days to establish a UUO model. **a** The immunoblot analysis of Col I&III, α-SMA, and DsbA-L. **b–e** Analysis of the grayscale image between them. **f** Immunohistochemical staining of Col I&III and α-SMA. **g–i** Quantification of immunohistochemical staining. Original magnification ×400. Scale bar:100 μM. Data are expressed as means ± s.d. ($n = 6$). #$P < 0.05$ versus sham group. *$P < 0.05$ versus PT-DsbA-L-WT with the UUO group. Each experiment (**a**, **f**) was repeated six times independently with similar results. **b–e**, **g–i** indicate the statistical Student's $t$ test used (means ± s.d., $n = 6$, $P < 0.05$).

Hsp90 with the mitochondria was similar to DsbA-L (see Supplementary Fig. S1D–F). Finally, an immunofluorescence confocal microscopy suggested that the colocalization signal of DsbA-L and Hsp90 in BUMPT cells, which had been treated with TGF-β1, as well as the kidneys of UUO mice and Ob patients, was more apparent in the BUMPT cells and kidneys of sham mice and Ob patients (see Supplementary Fig. S1G–I). Taken together, these data provided further proof to support the DsbA-L interaction with Hsp90 in the mitochondria.

**Suppression of Hsp90 attenuated TGF-β1-induced expression of Col I&III and vimentin via inactivation of Smad3 and down-regulation of p53 as well as of CTGF.** Previous research has shown that inhibition of Hsp90 by 17-dimethylaminoethyl amino-17-demethoxygeldanamycin (17-DMAG) attenuated TGF-β1-induced renal fibrosis via the inactivation of Smad2[20,24]. Our existing work demonstrated that deletion of tubular p53 attenuated renal fibrosis caused by TGF-β1 and UUO[11]. However, Hsp90-regulated expression of p53 depends on cell type and stimulation factor[25,26]. It is unclear whether Hsp90 positively regulates the expression of p53. Immunoblot results suggested that Hsp90β siRNA markedly attenuated the TGF-β1-induced expression of Col I&III, vimentin, and CTGF, as well as suppressing the activation of

Smad3 and p53, confirmed by the gray-level analysis (see Supplementary Fig. 2A–I). Collectively, these results indicate that Hsp90 mediated the TGF-β1-induced accumulation of ECM and CTGF expression via activation of Smad3 and p53.

**Incubation of mouse kidney fibroblasts and BUMPT cells transfected with DsbA-L results in an increase of ECM in mouse kidney fibroblasts via the production of CTGF.** To further investigate the mechanism of tubular cells drive the TIF, we co-cultured the cells of mouse kidney fibroblasts and BUMPT transfected with DsbA-L according to the description of "Methods." The results indicated that the supernatant of BUMPT cells transfected with DsbA-L significantly induced the expression of Col I&III and vimentin in mouse kidney fibroblasts, which was notably suppressed by BUMPT cells transfected with CTGF-neutralizing antibody (see Supplementary Fig. 3A–E). The data demonstrated that BUMPT cells transfected with DsbA-L produced CTGF, and CTGF subsequently promoted the accumulation of ECM in mouse kidney fibroblasts

**Inhibition of Hsp90 ameliorated UUO-induced renal fibrosis, renal cell apoptosis, and infiltration of F4/80.** Previous studies have reported that inhibition of Hsp90 suppressed UUO-induced

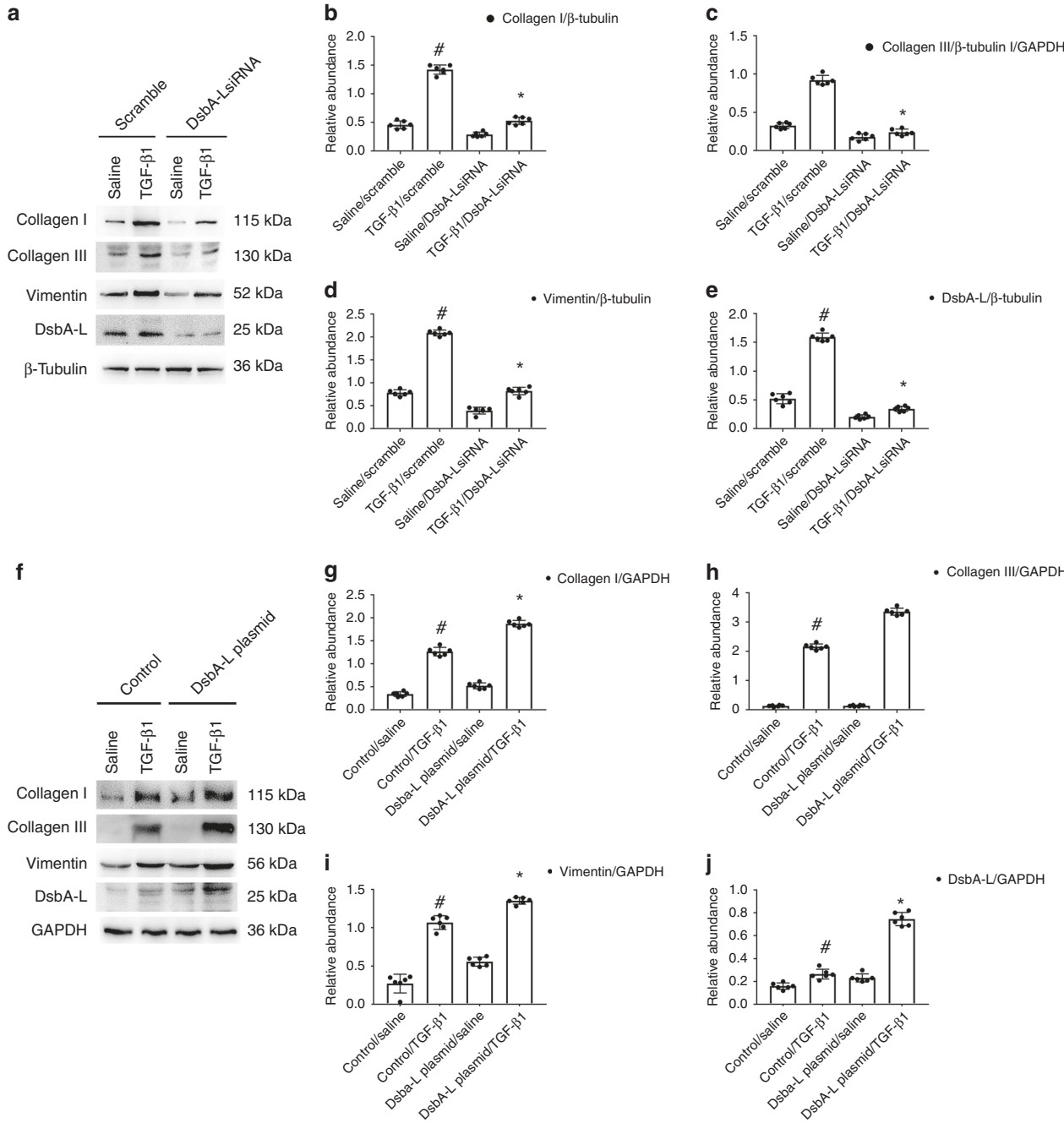

**Fig. 5 DsbA-L-mediated the TGF-β1-induced expression of Col I&III and vimentin in BUMPT cells.** The plasmid and siRNA of DabA-L were transfected into BUMPT cells and treated with or without 5 ng/ml TGF-β1 for 24 h. **a, f** Immunoblot analysis of Col I&III, vimentin, and DsbA-L. **b–e, g–j** Analysis of the gray scale image between them. Data are expressed as means ± s.d. ($n = 6$). #$P < 0.05$ versus scramble with the saline group. *$P < 0.05$ versus scramble with the TGF-β1 group. Each experiment (**a, f**) was repeated six times independently with similar results. **b–e, g–j** indicate the statistical Student's $t$ test used (means ± s.d., $n = 6$, $P < 0.05$).

renal fibrosis[20]. However, the effect of inhibition of Hsp90 for renal cell apoptosis and infiltration of F4/80 remains unclear. We found that Hsp90β siRNA significantly reduced the UUO-induced tubular dilation and atrophy (see Supplementary Fig. 4A). Masson's trichome staining results demonstrated the UUO-induced accumulation of ECM was notably reduced by Hsp90β treatment (see Supplementary Fig. 4B, E). TUNEL staining indicated that Hsp90β siRNA markedly attenuated the UUO-induced renal cell apoptosis (see Supplementary Fig. 4C). The TUNEL-positive cells further confirm the above findings (see Supplementary Fig. 4F). The UUO-induced infiltration of F4/80

macrophage cells into C57BL/6 mice kidney tissues was found to be significantly attenuated by Hsp90β siRNA treatment (see Supplementary Fig. 4D, G). Therefore, the data supported and extended the previous findings to demonstrate that the production of apoptosis and inflammation was related to the progression of renal fibrosis caused by Hsp90β.

**Attenuation of UUO-induced expression of Col I&III, α-SMA, CTGF, p-Smad3, and p53 in mice treated with Hsp90β siRNA.** Further verifying our in vitro findings, the immunoblot results demonstrated that Hsp90β siRNA notably attenuated the expression

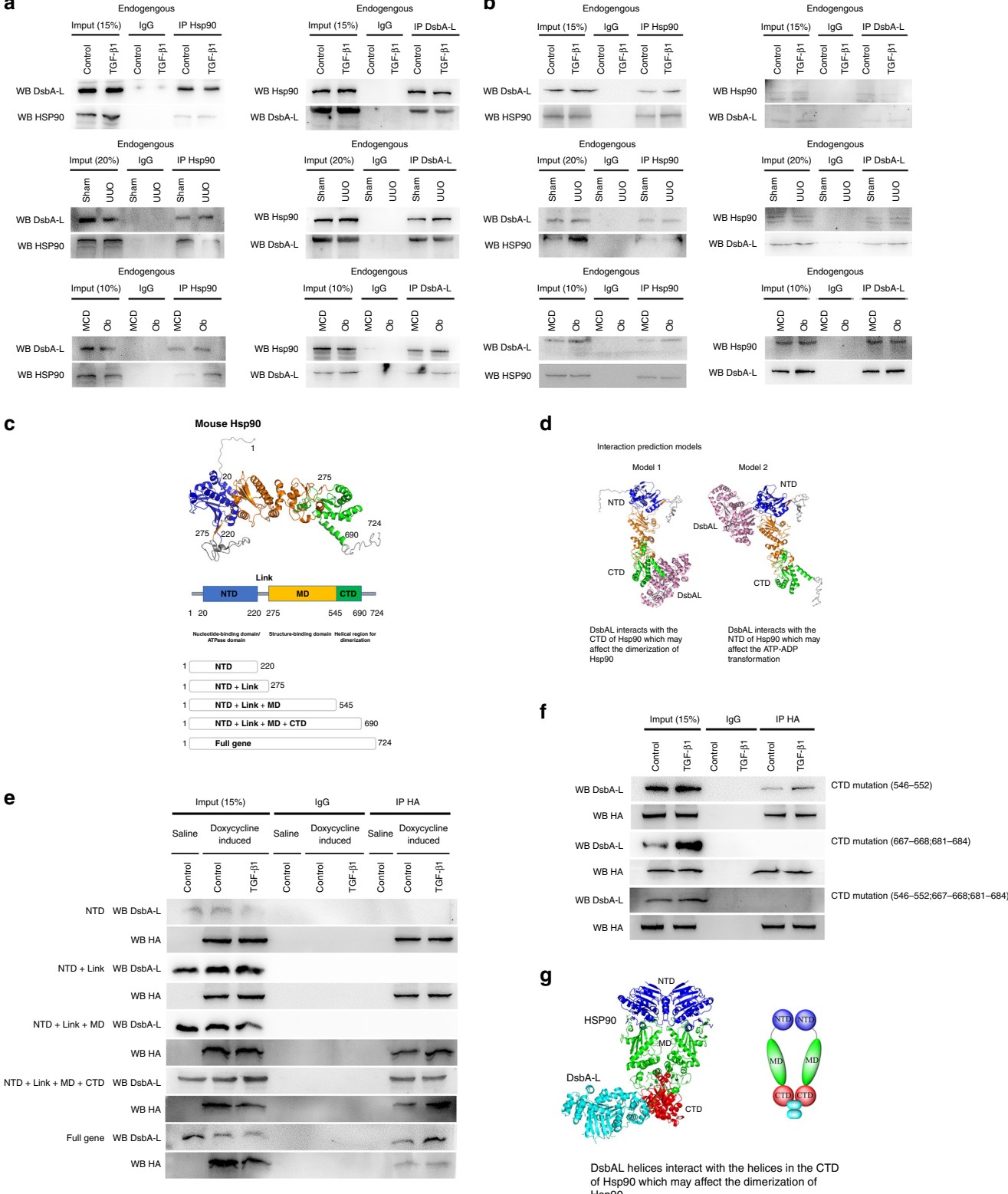

**Fig. 6 Interaction between DsbA-L and Hsp90 in BUMPT cells treated with TGF-β1 and the kidneys of UUO mice and Ob patients. a, b** The whole or mitochondrial lysate was extracted for reciprocal co-immunoprecipitation of DsbA-L and Hsp90 in BUMPT cells treated with TGF-β1 and the kidneys of UUO mice and Ob patients. **c** The structure and five sections of Hsp90. **d** Interaction of a predictive models. **e, f** Anti-HA immunoprecipitates were analyzed for HA and DsbA-L by immunoblotting. **g** The model of DsbA-L interaction with CTD of Hsp90β. Each experiment (**a, b, e, f**) was repeated six times independently with similar results.

of Col I&III, α-SMA, CTGF, p-Smad3, and p53 after UUO injury, as confirmed by the gray-level analysis (see Supplementary Fig. 5A–I). Immunohistochemistry staining further confirmed that UUO-induced accumulation of Col I&III and α-SMA was markedly reduced by Hsp90β siRNA treatment (see Supplementary Fig. 5J–M). Together, these in vivo data are consistent with the in vitro results demonstrating that Hsp90 mediated the UUO-induced renal fibrosis via activation of Smad3 as well as upregulation of p53 and CTGF.

**UUO-induced activation of Hsp90/Smad3/p53/CTGF axis was suppressed in PT-DsbA-L-KO mice.** In order to further confirm whether DabA-L regulates the Hsp90/Smad3/p53/CTGF axis, the littermate mice of PT-DsbA-L-WT and PT-DsbA-L-KO were subjected to UUO for 7 days. Immunoblot results indicated that UUO-induced Hsp90, p-Smad3, p53, and CTGF were markedly suppressed in PT-DsbA-L-KO mice (see Supplementary Fig. 6A). The gray-level analysis further verified immunohistochemistry results (see Supplementary Fig. 6B–F).

**Attenuation of UUO-induced TIF in PT-DsbA-L-KO mice was diminished by the overexpression of Hsp90.** To further confirm whether DsbA-L mediated the effect of TIF dependent on Hsp90, following the mice UUO model, Hsp90 plasmid was injected via tail veins. UUO-induced tubular dilation and atrophy were ameliorated in PT-DsbA-L-KO mice at day 7, which was deleted by the overexpression of Hsp90 (see Supplementary Fig. 7A). Masson's trichome staining results indicated that overexpression of Hsp90 eliminated the antifibrosis role of PT-DsbA-L-KO in the mice UUO model (see Supplementary Fig. 7A, B). Immunoblot results further confirmed that the reduction of UUO-induced expression of Col I&III and α-SMA in PT-DsbA-L-KO tissue was reversed by the overexpression of Hsp90 (see Supplementary Fig. 7C–H). Immunohistochemistry staining results also confirmed the finding of the immunoblot results (see Supplementary Fig. 7I–L). The data initially suggested that DsbA-L promoted TIF and relied on Hsp90.

**Aggravation of UUO-induced TIF caused by the overexpression of DsbA-L was eliminated by Hsp90β siRNA treatment.** While overexpression of Hsp90 eliminated the protective role of PT-DsbA-L-KO in UUO mice, it remains unclear whether inhibition of Hsp90 prevents DsbA-L-mediated TIF. Following the mouse UUO model, DsbA-L plasmid or Hsp90β siRNA was injected via a tail vein. Overexpression of DsbA-L aggravated UUO-induced tubular dilation and atrophy in all groups apart from the UUO mice, which had been treated with Hsp90β siRNA (see Supplementary Fig. 8A). Masson's trichome staining results indicated that overexpression of DsbA-L enhanced UUO-induced TIF apart from in the UUO mice, which had been treated with Hsp90β siRNA (see Supplementary Fig. 8A, B). Immunoblot results supported the findings of the immunohistochemistry staining, while DsbA-L was successfully transfected into the kidneys of the mice and Hsp90 was suppressed by Hsp90β siRNA treatment (see Supplementary Fig. 8C–H). Immunohistochemistry staining results also demonstrated that increasing UUO-induced expression of Col I&III and α-SMA in the overexpression of DsbA-L tissue was notably prevented by Hsp90β siRNA treatment (see Supplementary Fig. 8I–L). The data provided further evidence to suggest that DsbA-L-mediated TIF via Hsp90.

**PT-DsbA-L-KO mice attenuated the I/R, low-dose cisplatin, and aristolochic acid-induced renal fibrosis via suppression of Hsp90/ p53 and Smad3/CTGF/ axis.** Although we have demonstrated that PT-DsbA-L-KO ameliorated the UUO-induced renal fibrosis, it is not clear whether PT-DsbA-L-KO might attenuate the other tubular injury model (such as I/R, low-dose cisplatin, and aristolochic acid)-induced renal fibrosis. Interestingly, the data indicated that PT-DsbA-L-KO notably improved I/R, low-dose cisplatin, and aristolochic acid-induced renal function decline, tubular damage, and interstitial fibrosis, as demonstrated by the level of BUN and serum creatinine, hematoxylin and eosin (HE) and Masson's trichome stainings, tubular damage scores, and fibrotic area analysis (see Supplementary Figs. 9, 12, and 15). Furthermore, the immunoblot results indicated that PT-DsbA-L-KO also suppressed I/R, low-dose cisplatin, and aristolochic acid-induced expression of Col I&III and α-SMA via inhibition of Hsp90/p53 and Smad3/CTGF/ axis at indicated time points (see Supplementary Figs. 10, 13, 16, and 17). Finally, the immunohistochemistry staining showed that PT-DsbA-L-KO also suppressed I/R, low-dose cisplatin, and aristolochic acid-induced expression of Col I&III and α-SMA (see Supplementary Figs. 11, 14, and 18). Taken together, the data indicated that PT-KO-DsbA-L not only attenuated UUO-induced renal fibrosis but also ameliorated I/R, low-dose cisplatin, and aristolochic acid-induced renal fibrosis.

**Hsp90 siRNA attenuated the I/R, low-dose cisplatin, and aristolochic acid-induced renal fibrosis via suppression of p53 and Smad3/CTGF/ axis.** While inhibition of Hsp90 might reduce UUO-induced renal fibrosis, it is unclear whether Hsp90β siRNA might reduce I/R, low-dose cisplatin, and aristolochic acid-induced renal fibrosis. The data showed that I/R, low-dose cisplatin, and aristolochic acid-induced renal function deterioration, tubular damage, and interstitial fibrosis, which was markedly ameliorated by the Hsp90 siRNA (see Supplementary Figs. 19, 22, and 25). The immunoblot analysis demonstrated that I/R, low-dose cisplatin, and aristolochic acid induced the upregulation of Col I&III, α-SMA, Hsp90, p53, p-Smad3, and CTGF at indicated time points; this was reduced by the Hsp90β siRNA (see Supplementary Figs. 20, 23, 26, and 27). In addition, the immunohistochemistry analysis indicated that Hsp90β siRNA also suppressed I/R, low-dose cisplatin, and aristolochic acid-induced expression of Col I&III and α-SMA (see Supplementary Figs. 21, 24, and 28). Taken together, the data indicated that Hsp90 not only mediated UUO-induced renal fibrosis but also mediated I/R, low-dose cisplatin, and aristolochic acid-induced renal fibrosis.

**DsbA-L/Hsp 90/Smad3 and p53/CTGF axis in the human kidneys.** Although we have reported that DsbA-L was induced in Ob patients (Fig. 1), the expression of Hsp90/Smad3 and p53/ CTGF axis remains unclear. First, we showed the patients' basic information in Table 1. Furthermore, the results indicated that Ob induced the tubulointerstitial injury or fibrosis, and the expression of Hsp90, Col I&III, and α-SMA (see Supplementary Fig. 29A–E). Immunoblot results demonstrated that the expression of Hsp90, p-Smad3, p53, and CTGF was markedly induced in Ob patients (see Supplementary Fig. 29F–I). Combining the previous result, the data showed that DsbA-L/Hsp90/Smad3 and p53/CTGF axis was involved in the regulation of fibrosis.

## Discussion
Previous studies have demonstrated that overexpression of DsbA-L has a renal-protective role against lipid-related kidney damage in DN[17]. In the current study, we initially induced DsbA-L in TGF-β1-treated BUMPT cells as well as in the kidneys of UUO mice and Ob patients, especially as the mitochondrion damage existed in the UUO model[27,28]; we presumed that the expression of DsbA-L in kidney tubular injury was associated with the

**Table 1 The basic clinical information of MCD and Ob patients.**

| Characteristics | MCD (n = 8) | Ob (n = 8) | P value |
|---|---|---|---|
| Age (years) (mean) [s.d.] | 58.37 (4.00) | 51.62 (2.58) | 0.1782 |
| Gender | | | |
| Male | 6 | 4 | |
| Female | 2 | 4 | |
| BMI (mean) [s.d.] | 21.46 (0.71) | 21.68 (0.68) | 0.8318 |
| Degree of hydronephrosis | | | |
| 0 | 8 (100%) | 0 | |
| 1 | 0 | 0 | |
| 2 | 0 | 0 | |
| 3 | 0 | 3 (37.5%) | |
| 4 | 0 | 5 (62.5%) | |
| Diagnosis | Minimal change diseases | Hydronephrosis | |
| Operation side | | | |
| Left | 2 | 5 | |
| Right | 6 | 3 | |
| Blood urea nitrogen (mM) (mean) [s.d.] | 4.48 (0.60) | 4.11 (0.33) | 0.5945 |
| Serum creatinine (mM) (mean) [s.d.] | 75.63 (5.46) | 94.51 (4.42) | 0.0177 |

damage of mitochondrion damage (see Fig. 1 and Supplementary Fig. S1). To investigate the role of DsbA-Link kidney fibrosis, we generated proximal tubular-specific DsbA-L-KO mice. To our knowledge, we are the first to have found that tubular DsbA-L mediated UUO-induced TIF, renal cell apoptosis, and inflammation in mice. It is completely opposite to the role of DsbA-L in DN[17,18]; the possible reason is that the difference between the role of system and tubular KO DsbA-L in DN and UUO model. In addition, we also found that DsbA-L mediated I/R, low-dose cisplatin, and aristolochic acid-induced TIF. The data demonstrated that DabA-L mechanistically interacted with Hsp90 in the mitochondria and then activated both Smad3 and p53 to induce the expression of CTGF, hence leading to TIF. We have also examined consistent changes of these molecules in Ob patients (see Supplementary Fig. 29). The current study uncovered a pathway for DabA-L/Hsp90/Smad3 and p53/CTGF, which is responsible for the progression of TIF (see Supplementary Fig. 30). Together, these results suggest that inhibition of DsbA-L could prevent the progression of renal fibrosis.

DsbA-L tends to be involved in anti-inflammation, anti-insulin resistance, suppression of endoplasmic reticulum, and deletion of lipid droplet deposition[14–17]. However, the role of DsbA-L in fibrosis remains unknown. In the current study, data suggested that DsbA-L contributes to the progression of renal fibrosis as supported by the following evidence: (1) using tubular-specific deletion of DsbA-L mice, following UUO injury, TIF, renal cell apoptosis, and infiltration of F4/80 caused by UUO were also ameliorated in PT-DabA-L-KO (see Figs. 3 and 4). (2) PT-DabA-L-KO also attenuated TIF induced by the other tubular damage models, including the I/R, low-dose cisplatin, and aristolochic acid (see Supplementary Figs. 9–18). (3) Accumulation of ECM induced by TGF-β1 was reduced by DsbA-L, while enhancing the overexpression of DsbA-L (see Fig. 5). Collectively, these results provide strong evidence that DabA-L mediated the progression of renal fibrosis, and unexpectedly promoted renal cell apoptosis and inflammation.

Previous studies have demonstrated that Hsp90 was involved in apoptosis as well as inflammation in cardiac muscle and neuron cells[29,30]. Recent studies demonstrated that inhibition of Hsp90 by the 17-allylamino-17-demethoxygeldanamycin (17-AAG) or 17-DMAG attenuated UUO, indoxyl sulfate (IS), and high salt-diet-induced renal fibrosis[20,24,31]. Although 17-DMAG is better than 17-AAG[32] to void potential side effects of inhibitors, the Hsp90β siRNA was used in the present experiment. The results of the current study confirmed and extended previous findings to indicate that knockdown of Hsp90β using siRNA ameliorated TGF-β1-induced accumulation of ECM as well as the UUO, I/R, low-dose cisplatin, and aristolochic acid-induced TIF, renal cell apoptosis, and F4/80 infiltration (see Supplementary Figs. 4, 5, and 19–28).

Previous studies have also reported that inhibition of Hsp90 attenuated the TGF-β1/Smad2 signaling[20]. Smad2 and Smad3 are key members of the Smad family. However, their roles in fibrosis are different. One study reported that Smad2 had a protective role against fibrosis induced by TGF-β1[33]. Other studies have demonstrated that Smad3 mediated TGF-β1-induced renal fibrosis. We explored the TGF-β1/Smad3 signaling, and our results indicated that Hsp90β siRNA blocked the TGF-β1/Smad3 signaling both in vitro and in vivo. Our previous studies have found that deletion of p53 attenuated UUO, I/R-induced renal fibrosis, renal cell apoptosis, and inflammation[11,34,35]. Interestingly, the present results suggest that Hsp90β siRNA suppressed TGF-β1/p53 signaling. p53 was at least partially responsible for Hsp90-mediated UUO, I/R, low-dose cisplatin, and aristolochic acid-induced renal fibrosis, renal cell apoptosis, and inflammation. CTGF is a key profibrotic factor and inflammation producer[36–38]. It is worth noting that the current study demonstrated that Hsp90β siRNA suppressed the expression of CTGF, supported by the finding that CTGF induced by TGF-β1 was mediated by both Smad3 and p53[37,39]. In addition, the results of co-culture of BUMP cells transfected with DsbA-L and mouse kidney fibroblasts demonstrated that CTGF mediated the tubular DsbA-L-induced TIF (see Supplementary Fig. 3). Collectively, we suggest that Hsp90/Smad3 and p53/CTGF axis mediated UUO- or TGFβ1-mediated renal fibrosis.

DsbA-L-mediated, UUO-induced TIF was dependent on Hsp90. First, deletion of tubular DsbA-L notably suppressed the Hsp90/Smad3 and p53/CTGF axis following UUO injury (see Supplementary Fig. 6). Second, overexpression of Hsp90 eliminated the effects of PT-DsbA-L-KO and attenuated UUO-induced TIF (see Supplementary Fig. 7). Finally, overexpression of DsbA-L plasmid enhanced UUO-induced TIF; however, it did not increase TIF in UUO mice, which had been treated with Hsp90β siRNA (see Supplementary Fig. 8). Taken together, the data provide strong evidence that DsbA-L is induced by UUO via Hsp90. However, whether DsbA-L directly regulates Hsp90 remains unclear. Our co-IP analysis suggested that DsbA-L interacted with Hsp90 in BUMPT cells treated with TGF-β1 as well as in the kidneys of UUO mice and Ob patients (see Supplementary Fig. 1). Further research revealed that DsbA-L interacted with the 667–668;681–684 region of CTD of Hsp90β (see Fig. 6 and Supplementary Video), whether DsbA-L binds to other isoforms of HSP90α or TRAP1 requires further investigation, which was further confirmed by the co-localization of DsbA-L or Hsp90 with the mitochondria, and DsbA-L and Hsp90 both in vitro and in vivo (see Supplementary Fig. 1). Finally, taken together, the data suggested that DsbA-L interacted with Hsp90 to regulate TIF.

In conclusion, we demonstrated a direct causal link between DsbA-L expression in proximal tubular cells and TIF, evidenced by the attenuation of UUO, I/R, low-dose cisplatin, and aristolochic acid-induced TIF in proximal tubule-specific DsbA-L-KO mice. In BUMPT cells, the DsbA-L interacted with Hsp90 and activated both Smad3 and p53, resulting in the expression of CTGF, a key regulator of kidney fibrosis, and then promoted the increase of ECM in mouse kidney fibroblasts. Analysis of Ob

tissues suggested that the DsbA-L/Hsp90/pSmad3 and p53/CTGF axis might be involved in human kidney fibrosis. Our study suggests that this signal pathway may be a therapeutic target of organ fibrosis, which has been caused by the disease.

## Methods

**Antibodies and reagents**. Anti-GAPDH, anti-Hsp90, anti-β-actin, anti-Col I, anti-Col IV, anti-vimentin, anti-α-SMA, anti-fibronectin, and anti-CTGF antibodies were obtained from Abcam (Cambridge Science Park, Cambridge, UK). Anti-p53, p-Smad3, and Smad3 were purchased from Cell Signaling Technology (Danvers, MA, USA), while anti-COXIV and TGF-β were obtained from Proteintech (Rosemont, IL, USA). Anti-DsbA-L antibody was provided by Dr. Feng Liu Lab. All secondary antibodies, MitoTracker Green FM and MitoTracker Red CMXRos, were obtained from Thermo Fisher Scientific (Waltham, MA, USA). Recombinant human TGF-β1 was purchased from R&D Systems (Minneapolis, MN, USA). The cisplatin, aristolochic acid, and Mitochondria Isolation Kit were purchased from Sigma-Aldrich (Shanghai, China). The target sequence for mouse Hsp90β was described as in the previous study[40].

**Animals**. The proximal tubule-specific DabA-L-deletion mice were generated by crossing DsbA-L (flox/flox) mice (provided by Dr. Feng Liu Lab) with PEPCK-Cre mice (provided by Jackson Laboratory) as previously described[35]. The UUO model was established by ligating the left ureter in mice, also as previously described[11,41,42]. For ischemic acute kidney injury, it was induced by the duration of bilateral clamping for 28 min and followed by reperfusion[43]. For cisplatin injury, mice were intraperitoneally injected with 10 mg/kg cisplatin at weeks 0, 1, and 3[44]. For aristolochic acid injury, mice were intraperitoneally injected with 250 mg/kg aristolochic acid[45]. Hsp90β siRNA was administered twice a week by tail vein injection in C57BL/6 mice at a dose of 15 mg/kg, with saline as a control. Animal experiments were performed in accordance with guidelines approved by the Animal Care Ethics Committee of Second Xiangya Hospital, People's Republic of China, and ethical approval was obtained. Mice were housed in a 12-h light/dark cycle with free access to a standard rodent diet and water.

**Human samples**. After obtaining protocol approval from the Review Board of Second Xiangya Hospital, People's Republic of China, kidney biopsy specimens were obtained from patients living with MCD ($n = 8$) and Ob ($n = 8$). All MCD patients use biopsy as a clinical diagnostic standard. All organizations obtained the patient's consent. We declare that all study complies with all relevant ethical regulations for research with human participants and was carried out in compliance with the Declaration of Helsinki principles, and that the study is compliant with the guidance of the Ministry of Science and Technology for the Review and Approval of Human Genetic Resources.

*MCD*: Inclusion criteria: (1) pathologically diagnosed as a patient with clear minimal change diseases; (2) the patient does not have tumor and other kidney diseases; (3) age <75 years old. Exclusion criteria: (1) exclude patients with multiple renal tumors, central renal tumors, and other kidney diseases; (2) exclude patients with solitary kidneys (anatomically or functionally); (3) exclude patients whose blood creatinine is already abnormal in preoperative examination (male > 104.0 μmol/l; female > 87.1 μmol/l); exclude patients with other diseases (kidney stones, ureteral stones, etc.) who need other operations at the same time.

*Ob*: Inclusion criteria: (1) computed tomography confirmed severe unilateral renal hydronephrosis, (2) intravenous pyelography examination of hydronephrosis or not, but the opposite kidney was normal; (3) serum creatinine was normal; (4) postoperative medical examination was confirmed to be a pure hydronephrosis patient; (5) age <75 years old. Exclusion criteria: (1) exclude patients with tumors found in postoperative medical examinations, (2) exclude patients whose blood creatinine is already abnormal in preoperative examinations; (3) exclude patients with a history of diabetes, gout, hypertension, urinary tuberculosis, or infection. These were then used for HE staining, Masson's trichome staining, IP, and Western immunoblot staining.

**Cell culture, constructs, transfection, and treatments**. The BUMPT cells were cultured in DMEM (Dulbecco's modified Eagle's medium) (Thermo Fisher Scientific) with 10% fetal bovine serum, penicillin (100 U/ml), and streptomycin (100 μg/ml) in a humidified atmosphere of 5% $CO_2$ at 37 °C. Plasmids (HA-Dox-Teton-Hsp90β-NTD-1–220, HA-Dox-Teton-Hsp90β-NTD + Link-1–275, HA-Dox-Teton-Hsp90β-NTD + Link-1+MD-1–545, HA-Dox-Teton-Hsp90β-NTD + Link-1+MD + CTD-1–690, and HA-Dox-Teton-Hsp90β-1–724, HA-Hsp90β-Δ546–552, HA-Hsp90β-Δ545–690, and Hsp90β-Δ545–690) were transfected into BUMPT cells by lipofection. Twenty-four hours after transfection of DsbA-L siRNA, Hsp90β siRNA, negative control, or DsbA-L plasmid, BUMPT cells were subjected to starvation in a serum-free medium overnight and then were treated with or without TGF-β1 (5 ng/ml) for another 24 h, with 0.1% bovine serum albumin as a control. For the co-culture experiment, BUMPT cells were transfected with the plasmid of DsbA-L or control plus with or without CTGF-neutralizing antibody. After culturing with free serum for 24 h, the supernatant was collected and then transferred to culture with mouse kidney fibroblasts (Procell, CP-M069, Wuhan, China) for 24 h.

**Relative quantitative PCR (qPCR)**. Trizol reagent (Invitrogen, Carlsbad, CA, USA) was performed to extract total RNA from BUMPT cells, and kidney of C57BL/6J mice according to the manufacturer's procedure. Briefly, total RNA (40 ng) was reverse transcribed using M-MLV Reverse Transcriptase (Invitrogen). Real-time qPCR was carried out to detect the expression levels of DsbA-L and GAPDH using Bio-Rad (Hercules, CA) IQ SYBR Green Supermix with Opticon (MJ Research, Waltham, MA, USA) as previously described[12,46–49]. The primers used were as follows: DabA-L: 5′-AAATATGGGGCCTTTGGGCT-3′ (forward) and 5′-TAGCAACTCCAAGCGGTCAG-3′ (reverse); and GAPDH: 5′-GGTCTCCTCTGACTTCACA-3′ (forward) and 5′-GTGAGGGT CTCTCTCTTCCT-3′ (reverse).

**Protein-binding site prediction**. We used in silico approaches to identify potential functional domains that mediate the interaction between HSP90 and DsbAL. We used the sequence search in the RCSB PDB database to identify the PDB files of HSP90 dimer (ID: 2CG9) and DsbAL dimer (ID: 1YZX). Sequence analyses showed a high similarity between human and mouse sequences for HSP90 and DsbAL (see Supplementary Fig. S1). Second, given the extracted PDB as inputs, Cluspro V2.0 (http://cluspro.bu.edu/) was conducted to estimate structural interfaces that potentially mediate the protein interactions. A collection of 120 models was therefore obtained with the indication of potential interactions within the N-terminal domain, central domain, or C-terminal domain of HSP90. Third, we mapped the functional domains and performed cell-line experiments to validate the exact domains that mediated the HSP90–DsbAL interaction. Fourth, structural visualization and movies were prepared using PyMOL V2.1 (http://www.pymol.org/).

**Immunoprecipitation**. A Mitochondria Isolation Kit was used to isolate the mitochondria from BUMPT cells and kidney tissues following the standard procedure. The model of Hsp90 and DabA-L was described in Supplementary Methods. The IP buffer was used to lyse cells and was followed by IP using antibodies, including Hsp90, DsbA-L, and HA. Immunoprecipitates were resolved to sodium dodecyl sulfate-polyacrylamide gel electrophoresis (SDS-PAGE) and transferred to polyvinylidene difluoride membranes for immunoblot.

**Histology, immunohistochemistry, immunofluorescence, and immunoblot analyses**. Kidney tissues were harvested and observed using HE and Masson's trichrome stainings[41,50–52]. Immunohistochemical analyses were performed using anti-DsbA-L (1:50), Col I (1:100), Col III (1:100), and α-SMA (1:100) according to the previous protocol[41,42,53]. The immunofluorescence of DsbA-L and Hsp90 was carried out following the standard procedure[20,54–56]. Mitochondrial staining followed the standard operating procedure. The methods of quantitation have been described in our previous work[42]. Protein lysates from BUMPT cells or kidneys were subjected to SDS-PAGE and then to immunoblot analysis to detect the expression of DsbA-L, Hsp90, p-Smad3, Smad3, p53, Col I, Col III, α-SMA, TGF-β, and CTGF following the standard procedure.

**Statistics**. All data were presented as means ± s.d. Two group comparisons were made using two-tailed Student's $t$ tests. Multiple group comparisons were made using one-way analysis of variance. $P < 0.05$ was considered statistically significant.

**Reporting summary**. Further information on research design is available in the Nature Research Reporting Summary linked to this article.

## Data availability

All data in this study are true and reliable. The data that support the findings of this study are available from the corresponding author upon reasonable request. Source data are provided with this paper. The remaining figure data are uploaded as Supplementary information.

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

# Acknowledgements

This study was supported, in part, by a grant from National Natural Science Foundation of China [81870475, 81570646, and 81770951], Key R & D projects in Hunan Province [2018SK21215], Excellent Youth Foundation of Hunan Scientific Committee [2017JJ1035], and Changsha Science and Technology Bureau project [kq1901115]. We

would like to express our gratitude to EditSprings (https://www.editsprings.com/) for the expert linguistic services provided.

## Author contributions

D.Z. conceived and designed the experiments; X.L., J.P., and H.L. carried out the experiments; G.L., Z.H., Z.P., and H.Z. analyzed the data; Y.L., X.X., Y.Y., and F.L. contributed reagents/materials/analysis tools; D.Z. wrote the main manuscript text, but all authors reviewed the manuscript.

## Competing interests

The authors declare no competing interests.
