## [Peer Review File · Nature Communications]

Reviewers' Comments:

Reviewer #1:

Remarks to the Author:

In this manuscript Li, et al, propose a direct causal link between DsbA-L expression in proximal tubular cells and renal tubulointerstitial fibrosis (TIF). This is based on attenuation of unilateral ureteral obstruction (UUO)-induced TIF in proximal tubular specific DsbA-L-knockout mice. DsbA-L interacts with the C-domain of Hsp90 and appears to activate both Smad3 and p53, resulting in the expression of CTGF, a key regulator of kidney fibrosis. This in turn promotes the increase of ECM in NIH 3T3 cells. Analysis of tissues from obstructive nephropathy suggested that the DsbA-L/HSP90/pSmad3 and p53/CTGF axis might be involved in human kidney fibrosis.

This is an interesting topic but unfortunately the authors are present a story lacking key data. Additionally, key controls are missing, data are over-interoperated and they have not considered a simple explanation. It is really challenging to grasp the complexity of this study and how authors have made no attempts to present a simplified version. The manuscript is badly written and therefore it is very difficult to draw a conclusion based on poor quality of data presented here. These points are explained below.

Major comments:

- 1- Figure 1- A, B and C- Is DsbA-L upregulation at the mRNA level? Authors should present this data
- 2-Figure 1C- Why only one MCD sample presented?
- 3- Figure 1G- How long were BUMPT cells treated with TGF- β 1? Authors should perform an IF for same time points as in Figure 1B.
- 4- Figure 1I- Again, UUO sample is from which day? Tissues from different days (same as in figure 1A, should be stained.
- 5- Figure 1L- Is this really a representative of what was presented in Figure 1K? Authors need to stain the same tissues as in Figure 1C.
- 6- Figure 2C- Data presented here suggest that DsbA-L was not deleted in DsbA-L-KO mice. Please explain or provide additional data?
- 7- Figure 3C- TUNEL staining PT-DsbA-L-KO-UUO sample is not convincing. Staining should be repeated and background needs to be adjusted (lightened) to the same level as the other samples.
- 8- Figure 5A- Collagen I and Vimentin are still induced in TGF- β 1 siRNA . This does not agree with the conclusion.
- 9- Figure 6, 7- Does DsbA-L bind to Hsp90 alpha, beta or TRAP1?
- 10- Figure 8A-C lack controls. Alveospimycin (17-DMAG) treatment should lead to degradation of some bona fide client of Hsp90 such as AKT and phos-AKT. The fact that 17-DMAG does not enhance Hsp90 levels is concerning.
- 11-Figure 9B- The immunoblots are not convincing and need to be repeated
- 12-Figure 10- Is DsbA-La client of Hsp90? Another words treating the cells with 17-DMAG will lead to ubiquitination and degradation of DsbA-L.
- 13-17-DMAG is a very unstable drug. Authors should at least some of the experiments with 2nd generation of Hsp90 inhibitors. Or perhaps 17-AAG.

Minor comments-

- 1- Abstract- a simplified version of the abstract should be presented.
- 2- Figure 1G, I, K, 2E, 3ABCD, 4C, 7A-I, 10E need scale bars.
- 3- The manuscript needs an extensive editing and rewriting. For example;

"Line 312-313- The previous finding The data also demonstrated for the first time that DabA-L mechanistically interacted with HSP90 in"

"Line 318-320- . Together, these results suggest that a blockade of this pathway, such as by inhibiting DsbA-L, could prevent the progression of renal fibrosis."

4- Track changes on line 430-434 were not accepted.

Reviewer #2:

Remarks to the Author:

In this study the authors analyze the contribution of Disulfide-bond A oxidoreductase-like protein (DsbA-L) to tubulointerstitial fibrosis. they propose that loss of DsbA-L in proximal tubules protects the mice from UUO-induced TI fibrosis. At present the paper presents major concerns that overall dampen the enthusiasm for the study. They include

- 1) The rationale for looking at DsbA-L in kidney tubular injury is not sufficiently explained
- 2) The authors develop a mouse lacking DsbA-L in the proximal tubules, yet they perform an UUO model that affects collecting ducts. How do the authors explain the role of proximal tubules in this type of injury?
- 3) Proximal tubular injury models (more than one such as ischemia/reperfusion, aristolochic acid, cisplatin) need to be performed in order to clearly determine the contribution of proximal tubule DsbA-L in the progression of TIF.
- 4) The paper is of correlative evidence and quite overwhelming with too many pathways, too many conditions, too many cells types, and at present it is difficult to determine cause and effect.
- 5) The paper is poorly presented and poorly written. The words are not separated in certain instances and the captures of the figures are quite difficult to understand. The paper also contains grammatical errors.
- 6) the experiment in figure 8C is not clear. why the HSP90 inhibitor affects HSP90 expression?
- 7) The data in figure 7 (colocalization) are not convincing. While in the cells DsbA-L is expressed in the mitochondria, in the kidney tissues it seems to be expressed everywhere. The use of mice lacking DsbA-L in the proximal tubules should be used to check for autofluorescence vs. signaling.
- 8) The experiment with alvespimycin are difficult to understand and justify. A link between TGF- β and HSP90 has been established. To this end HSP90 has been shown to drive pulmonary fibrosis, high-salt induced nephropathy via activation of NF κ B and State. The use of HSP90 inhibitors has been shown to have a positive effect on fibrosis. So, it is not clear why the authors have decided to inhibit HSP90. Although it has a beneficial effect, it is not clear the relevance of this study to the DsbA-L.
- 9) The use of kidney derived cells including kidney fibroblasts should be considered. The authors use NIH 3T3 fibroblast that are known to express high levels of DsbA-L. However, a more physiological source of cells should be considered.

In conclusion, a better rationale for the study, a better injury model and a more organized and mechanistic study needs to be performed in order to better understand the role of DsbA-L in TIF.

December 11, 2019

Dear Dr. Francesco

Thank you for your efficient work in processing our manuscript entitled ““DsbA-L/HSP90/Smad3 and p53/CTGF axis-induced renal tubulointerstitial fibrosis in UUO mice” (Manuscript No: NCOMMS-19-23186A-Z).

After we have carefully read the comments from the reviewers, we realize that the major merits of our work were not fully identified or recognized by the reviewers.

Here we would emphasize that the most notable merits of our manuscript include:

- 1) We for the first time demonstrated that tubular DsbA-L-mediated UUO-induced TIF and renal cell apoptosis.
- 2) Mechanistically, the data also for the first time demonstrated that DabA-L mechanistically interacted with HSP90 in mitochondria and then activated both Smad3 and p53 to induce expression of CTGF, hence leading to TIF.
- 3) The above molecular mechanism was further verified using human Ob patients, which suggested that DsbA-L may be a potential therapy target for the fibrosis disease.
- 4) We have redone the related experiments according to the review comments.

I had opportunities to talk to Dr. Youhua Liu (Department of Pathology, University of Pittsburgh, Pittsburgh, PA, and Dr. Cijiang He (Division of Nephrology, Department of Medicine, Mount Sinai School of Medicine, New York, NY), and both read the manuscript and encourage me to write to you to ask if you could consider a revised version of the manuscript.

Point-by-point responses to the reviewers' comments are enclosed after the letter for your consideration.

In addition, this manuscript has been edited and proofread by the linguistic services expert EditSprings (<https://www.editsprings.com/>).

I will be most grateful if you could offer us a second opportunity

Look forward to hearing from you soon,

With kind regards,

Yours sincerely,

Dongshan Zhang

We would like to express our sincere thanks to the reviewers for the constructive and positive

comments.

Reviewer #1 (Remarks to the Author):

In this manuscript Li, et al, propose a direct causal link between DsbA-L expression in proximal tubular cells and renal tubulointerstitial fibrosis (TIF). This is based on attenuation of unilateral ureteral obstruction (UUO)-induced TIF in proximal tubular specific DsbA-L-knockout mice. DsbA-L interacts with the C-domain of Hsp90 and appears to activate both Smad3 and p53, resulting in the expression of CTGF, a key regulator of kidney fibrosis. This in turn promotes the increase of ECM in NIH 3T3 cells. Analysis of tissues from obstructive nephropathy suggested that the DsbA-L/HSP90/pSmad3 and p53/CTGF axis might be involved in human kidney fibrosis.

This is an interesting topic but unfortunately the authors are present a story lacking key data. Additionally, key controls are missing, data are over-interoperated and they have not considered a simple explanation. It is really challenging to grasp the complexity of this study and how authors have made no attempts to present a simplified version. The manuscript is badly written and therefore it is very difficult to draw a conclusion based on poor quality of data presented here. These points are explained below.

Major comments:

1- Figure 1- A, B and C- Is DsbA-L upregulation at the mRNA level? Authors should present this data

Response: I have redone it in Figure 1 A-C.

2-Figure 1C- Why only one MCD sample presented?

Response: Thank you very much for your suggestion, I have added two samples in Figure 1F

3- Figure 1G- How long were BUMPT cells treated with TGF- β 1? Authors should perform an IF for same time points as in Figure 1B.

Response: Thank you very much for your suggestion, I have done the staining of DsbA-L at 6h, 12h, and 24h after TGF- β 1 treatment.

4- Figure 1I- Again, UUO sample is from which day? Tissues from different days (same as in figure 1A, should be stained.

Response: Thank you very much for your suggestion, I have done the staining of DsbA-L at

6h, 12h, and 24h after TGF- β 1 treatment.

5- Figure 1L- Is this really a representative of what was presented in Figure 1K? Authors need to stain the same tissues as in Figure 1C.

Response: Thank you very much for your suggestion, I have done the staining of DsbA-L using the same tissues as in Figure 1 F(Old is Figure 1C).

6- Figure 2C- Data presented here suggest that DsbA-L was not deleted in DsbA-L-KO mice. Please explain or provide additional data?

Response: Thank you very much for your suggestion. The key reasons are as follows: The PT-DabA-L-Ko mice came from the DsbA-L flox/flox mice were crossed with PEPCEK-Cre mice, as Cre predominantly expressed in kidney proximal tubular cells(DS Zhang et al, J Am Soc Nephrol 2014, 25 (10), 2278-89), the DsbA-L in proximal tubular cells but not other kidney cells was deleted.

7- Figure 3C- TUNEL staining PT-DsbA-L-KO-UUO sample is not convincing. Staining should be repeated and background needs to be adjusted (lightened) to the same level as the other samples.

Response: Thank you very much for your suggestion, I have redone it, and asked you to review it.

8- Figure 5A- Collagen I and Vimentin are still induced in TGF- β 1 siRNA . This does not agree with the conclusion.

Response: Thank you very much for your suggestion, I have redone it, and asked you to review it in Figure 5A.

9- Figure 6, 7- Does DsbA-L bind to Hsp90 alpha, beta or TRAP1?

Response: Thank you very much for your question, DsbA-L binds to Hsp90 alpha.

10- Figure 8A-C lack controls. Alvepimycin (17-DMAG) treatment should lead to degradation of some bona fide client of Hsp90 such as AKT and phos-AKT. The fact that 17-DMAG does not enhance Hsp90 levels is concerning.

Response: Thank you very much for your question, I have replaced 17-DMAG with HSP90 siRNA to do related experiments, and asked you to review it.

11-Figure 9B- The immunoblots are not convincing and need to be repeated

Response: Thank you very much for your question, I have repeated it, and asked you to review it.

12-Figure 10- Is DsbA-L a client of Hsp90? Another words treating the cells with 17-DMAG will

lead to ubiquitination and degradation of DsbA-L.

Response: Thank you very much for your question, I replaced 17-DMAG with HSP90 siRNA to do related experiments, and did not find this effect.

13-17-DMAG is a very unstable drug. Authors should at least some of the experiments with 2nd generation of Hsp90 inhibitors. Or perhaps 17-AAG.

Response: Thank you very much for your question, I replaced 17-DMAG with HSP90 siRNA to do related experiments considering the potential negative effect of inhibitor, and asked you to review it.

Minor comments-

1- Abstract- a simplified version of the abstract should be presented.

Response: Thank you very much for your suggestion, I have done it, and asked you to review it.

2- Figure 1G, I, K, 2E, 3ABCD, 4C, 7A-I, 10E need scale bars.

Response: Thank you very much for your suggestion, I have done it, and asked you to review it.

3- The manuscript needs an extensive editing and rewriting. For example;

“Line 312-313- The previous finding the data also demonstrated for the first time that DabA-L mechanistically interacted with HSP90 in”

“Line 318-320- . Together, these results suggest that a blockade of this pathway, such as by inhibiting DsbA-L, could prevent the progression of renal fibrosis.”

Response: I have revised it. The paper has been edited by the linguistic services expert EditSprings (<https://www.editsprings.com/>), the edition certification has been uploaded as attached file.

4- Track changes on line 430-434 were not accepted.

Response: Thank you very much for your suggestion, I have revised it.

Reviewer #2 (Remarks to the Author):

In this study the authors analyze the contribution of Disulfide-bond A oxidoreductase-like protein (DsbA-L) to tubulointerstitial fibrosis. they propose that loss of DsbA-L in proximal tubules

protects the mice from UUO-induced TI fibrosis. At present the paper presents major concerns that overall dampen the enthusiasm for the study. They include

1) The rationale for looking at DsbA-L in kidney tubular injury is not sufficiently explained

Response: Thank you very much for your question, the data suggested that the mitochondrion damage existed in the UUO model (Mario Bianco, et al, PLoS One, 2019, 14 (6), e0218986; Kim SM, et al. Frontiers in immunology, 2018, 9, 2563). We also detected the DsbA-L mainly expressed in mitochondrion of kidney tubular, we presumed that the expression of DsbA-L in kidney tubular is associated with the mitochondrion damage.

2) The authors develop a mouse lacking DsbA-L in the proximal tubules, yet they perform an UUO model that affects collecting ducts. How do the authors explain the role of proximal tubules in this type of injury?

Response: Thank you very much for your question, I accepted that UUO model not only affects the collecting ducts, but also causes extensive proximal tubular degeneration (Forbes MS, et al, Am J Physiol Renal Physiol. 2012; Forbes MS, et al. Am J Physiol Renal Physiol. 2011). Furthermore, a lot of evidence has shown that the proximal tubules injury caused by UUO was involved in the renal tubulointerstitial fibrosis (Li H et al. Autophagy. (2016); Ma Z et al. Am J Physiol Renal Physiol. (2018); Mei S et al. Sci Rep. (2017); Li S, Mariappan N, et al. Am J Physiol Renal Physiol. 2013;). According to the previous study, we focused on the role of proximal tubules DsbA-L in UUO model using the lacking DsbA-L of the proximal tubules.

3) Proximal tubular injury models (more than one such as ischemia/reperfusion, aristolochic acid, cisplatin) need to be performed in order to clearly determine the contribution of proximal tubule DsbA-L in the progression of TIF.

Response: Your suggestion is so excellent, I accepted that I/R, aristolochic acid and repeated low-dose cisplatin-acute kidney injury (AKI) with acute tubules apoptosis or necrosis, and then gradually entered the chronic kidney disease (CKD) with renal fibrosis due to the over repair. The above models were usually used to study for transition of AKI to CKD. In current study, we only focused on the role of proximal tubules DsbA-L in TIF, hence, we selected the classical TIF model of UUO. According to your good suggestion, in future, we continued to investigate the role of proximal tubules DsbA-L in transition of AKI to CKD.

4) The paper is of correlative evidence and quite overwhelming with too many pathways, too many conditions, too many cells types, and at present it is difficult to determine cause and effect.

Response: Thank you very much for your good suggestion, our data indicated that TGF- β -induced DsbA-L, and then DsbA-L interacted with HSP90 in mitochondrion of kidney tubular, and subsequently activated both Smad3 and p53, resulting in the expression of CTGF, a key regulator of kidney fibrosis, and then promoted the increase of ECM in mouse kidney fibroblasts.

5) The paper is poorly presented and poorly written. The words are not separated in certain instances and the captures of the figures are quite difficult to understand. The paper also contains grammatical errors.

Response: I have revised it. The paper has been edited by the linguistic services expert EditSprings (<https://www.editsprings.com/>), the edition certification has been uploaded as attached file.

6) the experiment in figure 8C is not clear. why the HSP90 inhibitor affects HSP90 expression?

Response: Thank you very much for your question, I replaced 17-DMAG with HSP90 siRNA to do related experiments considering the potential negative effect of inhibitor, and asked you to review it.

7) The data in figure 7 (colocalization) are not convincing. While in the cells DsbA-L is expressed in the mitochondria, in the kidney tissues it seems to be expressed everywhere. The use of mice lacking DsbA-L in the proximal tubules should be used to check for autofluorescence vs. signaling.

Response: Thank you very much for your suggestion, I have redone it, and asked you to review them.

8) The experiment with alvespimycin are difficult to understand and justify. A link between TGF- β and HSP90 has been established. To this end HSP90 has been shown to drive pulmonary fibrosis, high-salt induced nephropathy via activation of NF κ B and Stat3. The use of HSP90 inhibitors has been shown to have a positive effect on fibrosis. So, it is not clear why the authors have decided to inhibit HSP90. Although it has a beneficial effect, it is not clear the relevance of this study to the DsbA-L.

Resonpes: Thank you very much for your question, although a link between TGF- β and HSP90 has been established, the role and regulation mechanism of HSP90 need to be further confirmed

and clarified in UUo model, respectively. In current study, I replaced 17-DMAG with HSP90 siRNA to do related experiments considering the potential negative effect of inhibitor. The data revealed that HSP90 siRNA attenuated the renal fibrosis via inactivation of Smad3 and p53/CTGF axis. Finally, we demonstrated that TGF- β -induced DsbA-L, and subsequently interacted with HSP90 in mitochondrion of kidney tubular, and then activated both Smad3 and p53, resulting in the expression of CTGF, a key regulator of kidney fibrosis, and then promoted the increase of ECM in mouse kidney fibroblasts.

9) The use of kidney derived cells including kidney fibroblasts should be considered. The authors use NIH 3T3 fibroblast that are known to express high levels of DsbA-L. However, a more physiological source of cells should be considered.

Response: Thank you very much for your good suggestion, I have redone it using the mice kidney fibroblats, and asked you to review it.

Reviewers' Comments:

Reviewer #1:

Remarks to the Author:

The authors have improved their manuscript significantly.
They have also addressed all of my comments and concerns.

However, they need to clarify the following points;

9- Figure 6, 7- Does DsbA-L bind to Hsp90 alpha, beta or TRAP1?

Response: Thank you for your question, as you raise a good point. DsbA-L binds to Hsp90 alpha.
- Which data supports this claim?

6- the experiment in figure 8C is not clear. why the HSP90 inhibitor affects HSP90 expression?

Response: Thank you for your question. Considering to the potential adverse effects of 17-DMAG on the expression of some proteins including the HSP90, thus, I have replaced 17-DMAG with HSP90 specific siRNA to carry out related experiments in Fig S2A&E (The original figure is Figure 8C). The data indicated that HSP 90 siRNA efficiently silenced the HSP90 expression.

-Which isoform of Hsp90 was knocked down?

Authors need to state this in their manuscript.

Reviewer #2:

Remarks to the Author:

the authors have taken into account all my major concerns and performed more in vivo models as well as in vitro experiments that clearly strengthen the significance of the study proposed and clearly support a role for this pathway in TI fibrosis.
overall the changes and additions are acceptable.

We would like to express our sincere thanks to the reviewer 1 for his constructive and positive comments that he provided concerning our manuscript.

Comments from reviewer 1

The authors have improved their manuscript significantly.

They have also addressed all of my comments and concerns.

However, they need to clarify the following points;

9- Figure 6, 7- Does DsbA-L bind to Hsp90 alpha, beta or TRAP1?

Response: Thank you for your question, as you raise a good point. DsbA-L binds to Hsp90 alpha.

- Which data supports this claim?

Response: Thank you very much for your help. The above answer is not complete. Actually, Heat shock protein HSP 90-beta, so-called HSP90ab1 (gene ID: 15516), was the interaction partner of DsbA-L according to our findings of bioinformatics, biochemical experiments, and mutagenesis analysis (Figure6). Whether DsbA-L binds to other isoforms of HSP90 alpha or TRAP1 requires further investigation.

6- the experiment in figure 8C is not clear. why the HSP90 inhibitor affects HSP90 expression?

Response: Thank you for your question. Considering to the potential adverse effects of 17-DMAG on the expression of some proteins including the HSP90, thus, I have replaced 17-DMAG with HSP90 specific siRNA to carry out related experiments in Fig S2A&E (The original figure is Figure 8C). The data indicated that HSP 90 siRNA efficiently silenced the HSP90 expression.

-Which isoform of Hsp90 was knocked down?

Response: Thank you very much for your question. We designed and synthesized a specific Hsp90-beta siRNA according the previous study(J Immunol, 178 (10), 6100-8 2007 May 15), hence, Hsp90-beta was knocked down.

Sincerely

Dongshan Zhang

Reviewers' Comments:

Reviewer #1:

Remarks to the Author:

The authors have addressed all my comments.